# Polysaccharide-Based Composite Films: Promising Biodegradable Food Packaging Materials

**DOI:** 10.3390/foods13223674

**Published:** 2024-11-18

**Authors:** Shengzi Li, Yu Ren, Yujie Hou, Qiping Zhan, Peng Jin, Yonghua Zheng, Zhengguo Wu

**Affiliations:** 1College of Food Science and Technology, Nanjing Agricultural University, Nanjing 210095, Chinaqipingzhan@njau.edu.cn (Q.Z.);; 2College of Food Science and Engineering, South China University of Technology, Tianhe District, Guangzhou 510640, China

**Keywords:** biomass polysaccharide, food packaging, preparation method, degradable material

## Abstract

With growing concerns about environmental protection and sustainable development, the development of new biodegradable food packaging materials has become a significant focus for the future of food packaging. Polysaccharides, such as cellulose, chitosan, and starch, are considered ideal biodegradable packaging materials due to their wide availability, good biocompatibility, and biodegradability. These materials have garnered extensive attention from researchers in food packaging, leading to considerable advancements in the application of polysaccharide-based food packaging films, coatings, aerogels, and other forms. Therefore, this review focuses on the application of polysaccharide-based packaging films in food storage and preservation and discusses their preparation methods, application progress, challenges, and future development directions. Through an in-depth analysis of the existing literature, this review aims to provide sustainable and environmentally friendly solutions for the food packaging industry.

## 1. Introduction

Food packaging films not only protect food from physical, chemical, and biological factors but also help extend shelf life and maintain food’s freshness and safety. Additionally, packaging films convey product information, enhancing consumers’ experiences [1]. Most of food packaging films currently in use are petroleum-based plastics, such as polyethylene (PE) and polypropylene (PP), which offer advantages like good mechanical properties, high barrier performance, practicality, and low cost [2,3]. However, due to their non-degradability and other shortcomings, the large-scale use of plastic packaging has led to the depletion of petroleum resources and environmental pollution, disrupting ecosystems and posing health risks to humans [4,5,6]. As a result, green, sustainable, and biodegradable bioplastics have become the ideal choice for developing a new generation of packaging films [7]. Compared to petroleum-based plastics, which take up to 500 years or more to degrade, these materials can decompose naturally within a shorter period (several months to years) after use, thus reducing the impact on the environment.

Researchers have increasingly focused on biomass resources, utilizing natural bio-based polymers to create biodegradable packaging films in place of traditional plastic ones. Natural biodegradable polymers, including polysaccharides (such as starch, cellulose, chitosan, sodium alginate, and hemicellulose), proteins (like soy protein and collagen), and lipids [8], are not only fully degradable but also inexpensive and biocompatible. Therefore, their use in food packaging films offers the advantages of being green, safe, and degradable. They not only provide physical protection but can also be modified to impart specific functionalities such as antimicrobial, antioxidant, and moisturizing properties [9].

Currently, polysaccharide-based packaging films are being used in the preparation of food preservation films, active packaging films, and smart packaging films for the transportation, storage and preservation of meat, fruits, and vegetables. However, polysaccharide-based packaging films still face several limitations and challenges in practical applications, such as cost issues, processing performance, and compatibility with existing packaging lines. In addition, packaging films made solely from natural polysaccharides have some drawbacks. For example, many natural polysaccharide-based packaging films exhibit moisture sensitivity, poor mechanical properties, susceptibility to microbial contamination, limited antimicrobial activity, and poor processability. These problems also limit the further development and large-scale application of polysaccharide-based packaging materials [10]. Researchers have conducted extensive studies, including the addition of metal nanoparticles, natural active ingredients [11], and the compounding of polysaccharide matrices with other polymers, to improve and enhance the performance of polysaccharide-based packaging films. This approach aims to meet the needs of a broader range of industrial and consumer applications, thus promoting their use in the food packaging industry.

Therefore, this review will introduce the preparation methods of packaging films and focus on the latest research advances in polysaccharide-based packaging films, discussing their applications, challenges, and future directions in food storage and preservation (Figure 1). Through an in-depth analysis of the existing literature, this review aims to provide sustainable and environmentally friendly solutions for the food packaging industry.

## 2. Preparation Methods of Polysaccharide-Based Packaging Films

There are various types of polysaccharide-based food packaging films, with differing preparation methods. Currently, the primary methods employed include casting, coating, electrospinning, and extrusion [12,13]. Each method is suited to specific materials, and the properties of the final films vary accordingly, making it crucial to select an appropriate preparation method.

### 2.1. Casting

The casting method involves homogenizing the components in an aqueous medium or organic solvent, followed by spreading the resulting dispersion onto a flat surface for drying to form a film layer (Figure 2a) [14]. Casting offers the advantage of producing uniform, clear, and strong films with simple equipment and high production efficiency [15], making it suitable for large-scale production. However, this method requires precise process control and high-quality raw materials to ensure the quality and safety of the final product.

The casting method has important applications in preparing food packaging films, meeting the diverse property requirements of these films. (1) Gas barrier: Food packaging films must possess good gas barrier properties to prevent the penetration of oxygen and moisture, thereby extending the shelf life of food. The casting method enables the creation of multilayer structures in polymer films, providing excellent barrier properties. (2) Mechanical strength: The mechanical strength of the film is the key to ensuring that the packaging material remains intact during transportation and storage. The casting method allows precise control over film thickness and structure, ensuring sufficient strength while maintaining flexibility. For example, Othman, SH et al. [16] developed corn starch/chitosan nanoparticles/thymol (CS/CNP/Thy) bio-nanocomposite films as potential food packaging materials that can enhance the shelf life of food. CS/CNP/Thy bio-nanocomposite films were prepared by the addition of different concentrations of thymol (0, 1.5, 3.0, 4.5 *w/w*%) using a solvent casting method. It was found that although the addition of thymol reduced the mechanical properties of the film, it was effective in lengthening the shelf life of cherry tomatoes, maintaining their firmness, reducing weight loss, and inhibiting *mold* growth by adjusting the concentration of thymol, especially at 3%. This suggests that CS/CNP/Thy bio-nanocomposite films have the potential to be applied in active food packaging to improve the preservation quality and prolong the shelf life of food.

### 2.2. Coating

Film coating is a method of forming a film by dipping or spraying film-forming components and additives into a liquid solution. As the water evaporates, a dense film forms on the surface of the food, acting as an insulator of O_2_, H_2_O, and CO_2_, thus reducing microbial-induced food spoilage.

Currently, the two most widely used coating methods are spraying and dipping [17]. Spraying is a conventional approach for applying low-viscosity coatings to food surfaces, typically using high-pressure atomization to produce fine droplets that deposit on the surface until a coating forms [18]. However, variations in operating conditions (e.g., droplet size, spray gun type, and pressure) can result in non-uniform coating thickness. Dipping is another method used to prepare coating solutions; however, the dipping method has its drawbacks. Firstly, dipping typically results in thick coatings, which may excessively reduce product respiration and damage the surface of the food, leading to degradation of functionality and reduction of storage properties. Secondly, cross-contamination can occur in the dipping solution. Finally, a large volume of coating solution is required per unit mass of the product to ensure proper impregnation. Therefore, impregnation is best suited for small-scale experiments, not for large-scale industrial production [19]. Electrospraying is a new technology for surface coating of food, in which the atomization of the coating solution occurs in a high-intensity electric field, allowing the formation of charged droplets with narrow dimensions [20]. When a high voltage is applied to the coating solution from the emitter tip, a Taylor cone forms, causing charges to accumulate on the droplet surface. This destabilizes the liquid, splitting it into numerous tiny charged droplets [21]. Due to electrostatic interactions, the coating film produced by electrospraying is more uniformly distributed than coatings applied with uncharged droplets [22]. For example, Ebru Ormanli et al. [23] used the electrospray method to coat a solution of fulvic acid (FA) and sericin (S) on paper to prepare active paper packaging materials (Figure 2d). They optimized the process by adjusting electrospray parameters to achieve minimal droplet size and maximize antimicrobial effects. In this study, FA and S with antioxidant and antimicrobial properties were coated directly onto packaging materials by electrospraying to extend the shelf life of pears.

Overall, the coating method is generally simple, easy to implement, and does not require complex equipment. However, spraying requires a safe, non-toxic film-forming solution, and uneven distribution in sprayed films limits the method’s broader applications.

### 2.3. Electrostatic Spinning

Electrostatic spinning utilizes electrohydrodynamics, where a polymer dispersion or melt is placed in a charged nozzle. When the applied voltage is sufficiently high, electric field forces stretch the polymer droplet into a Taylor cone. These forces overcome the surface tension, causing the droplet to be ejected as a filament. These filaments are evaporated by the solvent or cooled by the melt in real time and are eventually deposited on the receiver to form a nonwoven nanofiber network (Figure 2c) [14,24,25]. This technique is primarily used to create microfiber and nanofiber membranes from polymer dispersion or melts.

Electrostatic spinning can be further categorized into three distinct classes: uniaxial (co-mingled) electrostatic spinning based on spinning solutions or needles, coaxial electrostatic spinning, and emulsion electrostatic spinning. Uniaxial electrospinning, the most widely used technique, spins a homogeneous solution of the substrate and active materials through a single needle. In contrast, coaxial electrostatic spinning employs coaxial needles, which help to produce nanofibers with a “core–shell” structure. Finally, emulsion electrostatic spinning involves the direct spinning of stabilized emulsions (water-in-oil or oil-in-water emulsions), where the emulsion is mixed with the needle and then spun [26,27,28]. For example, Zhang et al. [29] designed a novel nanofiber membrane using coaxial electrostatic spinning, with thymol as the core layer and ethylene vinyl alcohol copolymer (EVOH) as the shell layer. The hydrophilic EVOH shell exhibited controlled release properties of thymol, which is responsive to changes in relative humidity, effectively extending the shelf life of strawberries. This provides efficient, environmentally friendly, and commercially viable solutions for food packaging. In 2024, Guan et al. [30] obtained cellulose nanocrystals (CNCs) from the extraction and bleaching of jute cellulose as the enhancer, 2-hydroxypropyl-β-cyclodextrin (HP-β-CD) as the carrier, and flavonoids—anthocyanidins and cinnamaldehyde—as the bioactive agent, and, finally, derived a novel kind of polylactic acid (PLA)-based composite membrane by the electrostatic spinning method (Figure 2b). The PLA-3.2 composite with tannic acid (TA) surface crosslinking demonstrated a 29.6% increase in mechanical strength over neat PLA, along with protective effects against oxidative stress and free radicals. In addition, this membrane had excellent cell biocompatibility, obvious intelligent color reaction, and good antibacterial ability. Finally, PLA-3.2 composites could be degraded by soil and are conducive to plant root growth. This work addresses current challenges related to the biodegradability and functionality of biopolymers, offering potential applications in intelligent, bioactive food packaging.

Electrostatic spinning technology for producing polysaccharide-based food packaging membranes offers several advantages, including high specific surface area and porosity, controllable fiber diameter, environmental friendliness, cost-effectiveness, functionality, and customizability. This technology can produce nanofiber membranes with specific functions, such as antimicrobial, antioxidant, and smart indicators, while ensuring biocompatibility and degradability, leading to low environmental impact and cost-effectiveness. This provides efficient, environmentally friendly, and commercially viable solutions in food packaging.

**Figure 2 foods-13-03674-f002:**
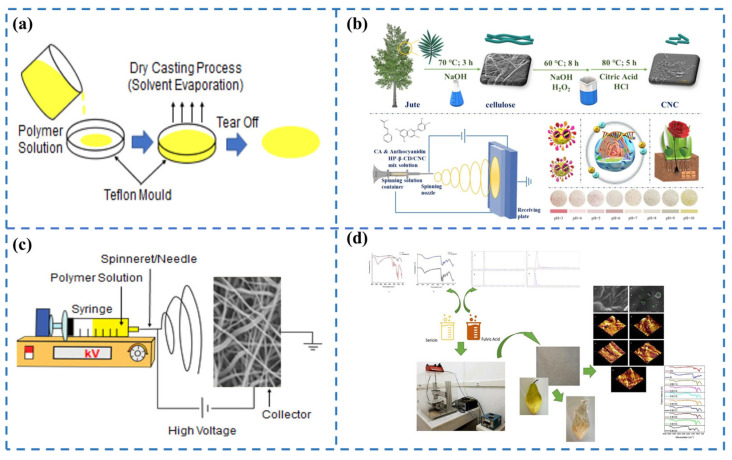
(**a**) Flow chart of polymer film preparation by the casting method [14]; (**b**) Fabrication route of CNC and PLA composites [30]; (**c**) Schematic diagram of the principle of the electrostatic spinning method [14]; (**d**) Paper packaging material prepared by the electrospray method to extend the shelf life of pears [23].

## 3. Applications for Polysaccharide-Based Food Packaging Film

Polysaccharides are macromolecular polymers composed of multiple monosaccharide units linked by glycosidic bonds. They occur widely in nature, including cellulose in plants, chitin in animals, and xanthan gum in microorganisms. Polysaccharide-based packaging films have gained considerable attention due to their renewability, biodegradability, and environmental friendliness. Moreover, the diverse chemical structures of polysaccharides allow for extensive modification, broadening their potential applications in food packaging.

### 3.1. Cellulose

Cellulose, the most abundant and renewable natural organic macromolecule, is derived from a wide range of sources. Structurally, cellulose consists of D-glucopyranosyl units linked by β-(1,4)-glycosidic bonds, with each glucose unit containing hydroxyl groups on C_2_, C_3_, and C_6_. These hydroxyl groups enable hydrogen bonding within and between cellulose molecules, leading to the formation of stable crystalline structures. As a non-toxic and renewable material, cellulose is valued as an eco-friendly resource due to its broad availability and environmental compatibility. Composite materials composed of cellulose and nanoparticles, such as nanosilver [31], nano zinc oxide [32], and nano copper oxide [33]), exhibit excellent antimicrobial properties, making them highly suitable for food packaging and related applications. Compared with traditional food packaging materials such as polyethylene (PE), cellulose-based materials with antimicrobial properties can effectively inhibit the growth of bacteria in food, extending the shelf life of food, and are biodegradable in the natural environment, thus contributing to environmental protection.

In recent years, increasing numbers of researchers have shown interest in composites prepared from cellulose nanofibers or recycled cellulose films with certain organic, inorganic, or natural additives such as nanosilver, nano silicon dioxide, chitosan, and plant essential oils, which enhance the mechanical strength of the material while providing barrier and antimicrobial effects. For example, to address the shortcomings of cellulose films in terms of insufficient barrier and antimicrobial properties, researchers made broadleaf cellulose as a raw material and employed the N-methyl morpholine-N-oxide method to mix the cellulose film with inorganic materials, such as nano silicon dioxide (SiO_2_), mesoporous silicon dioxide, nano sodium carbonate, nanosilver-carrying titanium dioxide, and silver-carrying mesoporous silicon dioxide. This research addressed the problems of low oxygen permeability, high moisture permeability, and poor strength of cellulose film, which has provided ideas for the development of fresh packaging film that can replace traditional plastics. In 2019, Wu et al. [34] prepared a novel multifunctional film (TNC/GSE/AgNPs) composed of TEMPO-oxidized nanocellulose (TNC), grape seed extract (GSE), and TNC-immobilized silver nanoparticles (TNC@AgNPs) (Figure 3a). Compared to pure TNC films, TNC/GSE/AgNPs films demonstrated superior mechanical properties, lower water vapor permeability, and reduced oxygen permeability. In addition, the film showed strong antioxidant and antimicrobial activity. Achieving super mechanical strength and high barrier properties remains a significant challenge. In 2023, Tang et al. [35] prepared superhydrophobic composite membranes with a sandwich-like structure using superhydrophobic MXene and TEMPO-oxidized nanocellulose using a layer-by-layer self-assembly method (Figure 3b). The resulting hybrid film showcased exceptional flexibility and strength, with the ability to be folded into various shapes and withstand significant weight (about 1 kg). The composite film exhibited superhydrophobicity, high photothermal conversion efficiency, and stability. Interestingly, the combination of the two succeeded in achieving controlled light-driven motion and enhanced antimicrobial properties, along with superhydrophobicity, anti-adhesion, and long-lasting photothermal sterilization properties. In the same year, Tang et al. [36] also developed a flexible gas barrier membrane with a nacre-like layered structure. This design utilized 1D TEMPO-oxidized nanofibrillar cellulose (TNF) and 2D MXene in an intertwined stacked arrangement, with OD AgNPs filling the gaps (Figure 3c). The strong interactions and dense structure give TNF/MX/AgNPs films mechanical properties and acid–base stability far superior to those of PE films. Importantly, the film exhibits ultra-low oxygen permeability and superior barrier properties to volatile organic gases, as confirmed by molecular dynamics simulations. In 2024, Wang et al. [37] obtained high-strength, high-barrier, and controlled sterilization nanocellulose-based bioplastic packaging (CTa-Ag) by immobilizing prepared Ta_4_C_3_T_x_ with AgNPs as an immobilization template, using an situ method, quaternized chitosan (QCS) as a green reducing agent, AgNPs as a reinforcing filler, and nanocellulose as substrate. Meanwhile, CTa-Ag has a unique photothermal conversion effect for the controlled release of antimicrobial active factors while immobilizing AgNPs to reduce their cumulative toxicity (Figure 3d). Regarding smart packaging films, Zhou et al. [38] developed a smart packaging for meat preservation and freshness detection by mixing different ratios of curcumin and anthocyanin into bacterial cellulose/gelatin-based films. Over time, the film’s color changed from yellow to red, with the film containing a 5:5 doping ratio demonstrating the best mechanical properties. The results show that the film can be used for both meat preservation and freshness monitoring. Therefore, cellulose holds great promise in food packaging film.

### 3.2. Chitosan

Chitosan, derived from the deacetylation of chitin, is the second most abundant macromolecule in nature after cellulose, and contains no artificial ingredients or drugs and poses no potential harm. Chemically, it is known as (1-4)-2-Amino-2-deoxy-β-D-glucan [39,40]. Chitosan is widely distributed in nature and has excellent film-forming and antibacterial properties. It is often referred to as the “sixth life factor” for the human body and has been recognized as “human-friendly” and “allergy-free” food. It is one of the few negatively charged natural substances capable of forming colloids and is the only known natural alkaline polysaccharide. In a mildly acidic environment, chitosan dissolves and forms a film.

Chitosan and its derivatives possess many advantages and beneficial properties in the field of food packaging films. For instance, they are frequently used as antibacterial agents, which can inhibit the growth and reproduction of certain microorganisms (fungi, bacteria, and viruses), and have good activity in antimicrobials. Existing studies have demonstrated three potential antimicrobial properties of chitosan: first, its ability to affect the rheological and osmotic properties of pathogenic bacterial cell membranes through the nature of the polycation; second, its ability to block DNA synthesis and expression; third, its ability to prevent and terminate the metabolic processes of pathogenic bacteria. When applied to food packaging, chitosan forms a tight film on the food’s surface, effectively isolating it and inhibiting bacterial nutrient absorption. Thus, chitosan-based biomaterials create a protective barrier that enhances food stability, extends shelf life, and helps preserve the original flavor of food products.

Research on chitosan-based biodegradable food packaging film dates back to the 1980s and 1990s, spanning three to four decades of development. Kittur et al. [41] prepared pure chitosan films using the casting method. The films exhibited moderate water vapor permeability, good oxygen barrier properties, and strong mechanical strength. However, pure chitosan films exhibit insufficient antimicrobial and antioxidant capacities, which limits their application in maintaining food quality and extending shelf life. To address this issue, researchers are exploring the addition of antimicrobial and antioxidant components to the films to improve the preservative and freshness retention capabilities of packaging films. Plant essential oils are frequently used to enhance chitosan films due to their excellent antimicrobial and antioxidant properties. Studies have shown that adding plant essential oils to chitosan matrices not only enhances the physical and barrier properties of the films but also effectively scavenges free radicals and inhibits the growth of microorganisms on the surface of the food, preventing food spoilage.

As a result, numerous studies have explored the incorporation of essential oils into chitosan matrices to prepare thin films. For example, Tulsi essential oil (TEO), rich in the antioxidant compound eugenol, was added to chitosan films. The resulting chitosan/TEO films demonstrated reduced water solubility and water vapor transport rates. Moreover, DPPH and H_2_O_2_ scavenging tests confirmed the excellent antioxidant properties of the film, and its application to French fries packaging also demonstrated its good anti-oil oxidizing activity [42]. Park et al. [43] first prepared capsules containing clove bract essential oil encapsulated with chitosan and carrageenan, and then added these capsules along with red kale pigment to the chitosan film. The results showed that the composite film significantly inhibited the growth of *fish spoilage bacteria* and changed color from purple to dark blue as the spoilage bacteria grew, clearly indicating fish freshness. Besides essential oils, metal nanoparticles exhibit broad-spectrum antimicrobial properties that can kill most bacteria, fungi, molds, spores, and other microorganisms upon contact, with a long-lasting antimicrobial effect [44]. Compared to plant essential oils, their antimicrobial properties are superior, more comprehensive, stable, and easier to synthesize. Metal nanoparticles currently used in food packaging include nanocopper [45], nanosilver [46], and some metal oxides such as zinc oxide nanoparticles [47] and nano titanium dioxide [48], which are often added to the film-forming dispersion to enhance the properties of the chitosan-based film, such as the mechanical, barrier, thermal, optical, and antimicrobial properties. Among them, silver nanoparticles are an excellent antimicrobial agent but have limited applications in the food sector due to their tendency to leak. In 2018, Wu et al. [49] synthesized labradorite-immobilized silver nanoparticles (LAP@AgNPs) as a green reducing agent using quaternized chitosan, in which AgNPs were embedded in the interlayer structure of labradorite due to confinement effects. Subsequently, chitosan-based films (Figure 4a) were preserved with LAP@AgNPs to keep lychee fresh and effectively extend its shelf life. This study offers a potential solution for the utilization of silver nanoparticles in the food sector. However, the cumulative toxicity of AgNPs limits their use in food packaging. Therefore, the pursuit of AgNPs should focus on controlling their release to minimize cumulative toxicity. In 2021, Wu et al. [50] treated sulfhydryl-modified chitosan using two green hydrothermal carbonization methods to obtain two types of carbon spheres/AgNPs (glutinous rice and sesame seed sphere-like AgNPs-SMCS and dragon fruit-like SMCS-Ag), which showed excellent stability and high immobilization efficiency for AgNPs. After 14 days, the total Ag release from AgNPs-SMCS and SMCS-Ag was only around 5.63% and 3.59%, respectively. Subsequently, Wu et al. incorporated each of these components into chitosan to prepare chitosan-based films (Figure 4c). Due to electrostatic interactions and micron filling behavior, the two carbon spheres/AgNPs modulate the microstructure of the chitosan-based film, further enhancing the immobilization of AgNPs. Importantly, these films have good antimicrobial activity and excellent safety. These findings provide a theoretical basis for the green and safe design of AgNP-based antimicrobial agents.

In addition to incorporating plant-derived active ingredients and metal nanoparticles as chitosan film enhancers, blending synthetic macromolecules or natural polymers into the chitosan matrix is also an effective approach to overcoming these shortcomings and developing food packaging [51]. For example, Zhou et al. [52] incorporated bacterial cellulose and tea polyphenols into the chitosan matrix to create biodegradable films (Figure 4d), significantly improving water vapor barrier, antioxidant, and antimicrobial properties, which extended the shelf life of grass carp. Haghighi et al. [53] produced a blend of chitosan and poly(vinyl alcohol) with the addition of ethyl laurate. The incorporation of PVA enhanced the mechanical properties and stability of the chitosan film, while ethyl laurate improved UV and light-barrier capabilities and significantly increased antimicrobial activity, making the composite film an eco-friendly antimicrobial packaging material. Ji et al. [54] prepared a biodegradable chitosan-based film containing micro ramie fiber and lignin by the casting method (Figure 4b). This was the first application of ramie fiber to reinforce chitosan films. The addition of these reinforcements notably enhanced mechanical properties, water resistance, thermal stability, and antioxidant activity. The addition of 20 wt% ramie fiber increased the tensile strength by 29.6%, while 20 wt% lignin increased antioxidant activity by 288% and reduced water absorption by 41.2%, and they both reinforced the thermal stability of the film. The films also demonstrated improved food preservation effects compared to polyethylene films, showing great potential for use in food packaging.

In recent years, significant progress has been made in the design and research of chitosan-based food packaging films. The modification of chitosan, along with the selection of suitable natural active ingredients, metal nanoparticles, or other polymers to compound with chitosan, aims to develop chitosan-based food packaging materials with high mechanical strength and potent antimicrobial and antioxidant properties. Developing such chitosan-based packaging materials remains a primary focus of current research efforts.

**Figure 4 foods-13-03674-f004:**
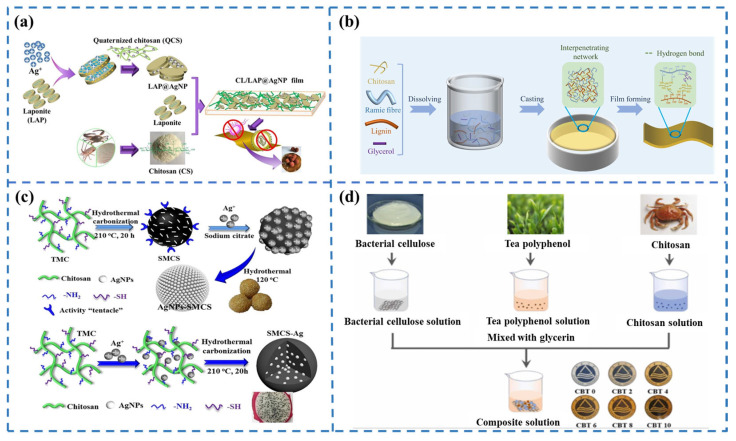
(**a**) Chitosan complexed with LAP@AgNP to prepare thin films [49]; (**b**) Schematic representation of the strategy for film preparation [54]; (**c**) Preparation of antimicrobial agents in the form of glutinous rice grains and dragon fruit by immobilization of nanosilver on carbon spheres [50]; (**d**) Bacterial cellulose and chitosan composite films [52].

### 3.3. Starch

Starch is a macromolecular carbohydrate widely found in plants and crops such as wheat, corn, rice, beans, and potatoes. It is safe, non-toxic, biodegradable, and exhibits good film-forming properties, making it a promising material for eco-friendly packaging. Starch, the main form of stored carbohydrates in plant cells, is composed primarily of amylose (straight-chain starch) and amylopectin (branched-chain starch). Both are polymers of α-D-glucose linked by glycosidic bonds, differing in their structural linkages, though both share the molecular formula (C_6_H_12_O_6_).

The primary goal of creating eco-friendly packaging materials made from starch that can be digested by the surrounding ecosystem is to facilitate resource recycling by converting the material into organic waste. One unique advantage of this plant-based material is its ability to withstand high temperatures without losing shape. In addition, these sheets have an appropriate level of suppression of gas permeability. However, starch-based packaging materials lack sufficient strength and water erosion resistance, resulting in weak physical properties, and their antibacterial and antioxidant capacities are low, making single starch-based packaging insufficient to meet food packaging requirements. Therefore, to improve the performance of starch-based materials, researchers have modified and enhanced them to meet various application needs. First, adding antibacterial substances to starch-based materials can significantly enhance the antibacterial ability of packaging materials. In 2021, Wu et al. [55] added various concentrations of lauryl arginine ethyl ester (LAE) to a mixture of polyvinyl alcohol (PVA) and starch, resulting in enhanced antimicrobial and physical properties of the film. These PVA/starch/LAE hybrid films exhibited good antibacterial property against *Escherichia coli* and * Staphylococcus aureus*. As the LAE concentration increased, the antibacterial activity of the films improved, but their transparency and mechanical properties decreased. In addition, recent studies have demonstrated that incorporating chitosan nanoparticles (CNPs) into starch-based films enables the fabrication of starch/CNPs bio-nanocomposite films. For instance, Lee et al. [56] successfully prepared coumaric acid-modified chitosan/chitosan nanoparticles/polyvinyl alcohol/starch composite films. The composite film not only possessed antimicrobial and antioxidant activities (Figure 5a) but also significantly enhanced mechanical, thermal, and barrier properties. On this basis, smart color-developing packaging films can be prepared by adding pH-sensitive color developers. In 2019, Qin et al. [57] developed a pH-sensitive packaging film based on tapioca starch and Lactobacillus helveticus anthocyanins (LRA). Incorporating anthocyanins into this packaging film significantly improved the UV–visible barrier capacity, tensile strength, and antioxidant potential without affecting its thermal stability. The film was used for pork freshness monitoring and showed good color development (Figure 5b). Subsequently, Li et al. [58] developed a smart active food packaging film based on polyvinyl alcohol (PVA) and tapioca starch using lactic acid ethyl ester acid arginine (LAE) as an antimicrobial agent and mulberry anthocyanins as a pH indicator. The results showed that the mechanical properties, ultraviolet light barrier, water vapor transmission rate, and moisture absorption of the films changed with the increase of anthocyanin content. In particular, the film containing 5% LAE and 5% mulberry anthocyanins effectively inhibited the growth of Escherichia coli and Staphylococcus aureus, and slowed down the milk deterioration process, demonstrating potential as a smart food packaging material.

Direct chemical modification of starch is also an effective method to enhance the performance of starch-based food packaging films. Additionally, properties of these films can be improved through techniques such as amination, increasing amylose content, and utilizing ultra-micronization technology. For example, Gurler et al. [59] modified potato starch (PPS) purified from waste potatoes by 3-(aminopropyl) trimethoxysilane (3-APTMS) to prepare crosslinked thin films, which were then combined with polylactic acid (PLA) by casting to prepare bilayered PPS-3APTMS-PLA films (Figure 5c). The silane-doped bilayer films demonstrated superior thermal stability, along with improved mechanical properties, water vapor permeability, optical clarity, and biodegradability compared to unmodified PPS. In biodegradation tests under simulated composting conditions, PPS, PPS-3APTMS, and PPS-3APTMS-PLA films showed biodegradation rates of 9.30%, 5.45%, and 5.08%, respectively. This study also examined the films’ microstructure, thermal properties, morphology, and nuclear magnetic resonance (NMR) spectra, suggesting a viable approach for developing food packaging and coating materials from waste starch.

In addition, resistant starch with a high amylose content is increasingly applied to food packaging, which helps overcome the limitations of natural starch, such as poor mechanical properties. Zou et al. [60] developed glycerol-plasticized composite films based on high-amylose corn starch (HCS) and konjac glucomannan (KGM) (Figure 5d), which showed a more homogeneous texture in the films’ micromorphology, and significantly improved the tensile strength, elongation at break, and water resistance of the HCS films. However, starch has a smaller crystalline region and is more susceptible to chemical degradation than cellulose, and its denaturation process carries a cost. Additionally, starch is more difficult to degrade after modification than cellulose, making it essential to maintain appropriate temperatures during processing to avoid excessive depolymerization and loss of functional properties.

### 3.4. Other Polysaccharides

Alginate was approved by the U.S. Food and Drug Administration in 1983 for use as a food additive ingredient, and today, it is widely applied in the food, pharmaceutical, environmental, and packaging industries. Derived primarily from marine brown algae, alginate is a natural hydrophilic polysaccharide biopolymer with excellent gel-forming properties, often utilized in hydrogel preparation. Available in the form of films or coatings, it offers good film-forming capabilities, low permeability to oxygen and water vapor, flexibility, water solubility, and a glossy, tasteless, and odorless surface.

When sodium alginate is combined with additives such as organic acids, essential oils, plant extracts, bacteriocins, or nanomaterials, it enhances food moisture retention, reduces shrinkage, delays oxidation, inhibits color and texture degradation, and lowers microbial load. For example, Sen et al. [61] prepared polyelectrolyte-structured antimicrobial food packaging films using starch, cationic starch, and sodium alginate, without using any conventional antimicrobial agents. These films displayed strong thermal stability, antimicrobial efficacy, and favorable surface properties, indicating suitability for various industrial applications. Luo et al. [62] added water extract (WE) and ethanol extract (EE) from guava leaves to sodium alginate to make a novel bioactive film. Compared to pure sodium alginate film, the guava leaf extract-enhanced film showed significantly improved antioxidant activity, antimicrobial effectiveness, tensile strength, and water barrier properties, along with reduced moisture content and elongation at break. FTIR and SEM analyses revealed that intermolecular hydrogen bonding between the guava leaf extract and the sodium alginate resulted in a more compact composite film structure. These findings suggest that sodium alginate–guava leaf extract films hold promise as antioxidant and antimicrobial food packaging materials.

Konjac glucomannan (KGM) is a non-toxic, biocompatible polysaccharide widely applied in food additives, food packaging, and biomedical fields [63]. However, KGM-based films lack mechanical strength and gas barrier properties due to their high hydrophilicity, limiting their practical application in food packaging [64]. To improve the usability of KGM-based films, methods have been explored to enhance film functionality by incorporating functional fillers like plant essential oils [65], plant polyphenols [66], and titanium dioxide [67]. In addition, other methods like blending with biopolymers [60], using emulsion-based film systems [68] or multilayer film systems [69], and employing electrostatic and microfluidic spinning techniques [70] can be utilized. Liu et al. [68] successfully developed edible films based on KGM with different content of bacterial cellulose nanofibers (BCNs). The study found that the surface roughness, mechanical strength, thermal stability, and barrier properties of the films were improved with the increase in BCN content. In particular, KGM films containing 2% BCNs exhibited the best performance, which suggests that BCNs, as a reinforcing agent, hold great potential for enhancing the usability of KGM-based films in food packaging.

## 4. Summary and Outlook

Currently, food production faces numerous challenges, particularly in subtropical regions, where high-quality fruit production is characterized by regional and seasonal factors. Freshly harvested fruits exhibit high physiological activity and moisture content, making them highly perishable. Additionally, concerns about pesticide residues and latent bacteria raise food safety issues for consumers. The absence of adequate preservation technologies and materials presents major challenges to the storage, transportation, and sale of fruits, vegetables, and other perishable food products. Consequently, as renewable resources, natural polysaccharides have attracted much attention in food preservation. This paper reviews the application of polysaccharides in food packaging and systematically examines the characteristics of polysaccharide-based materials and their application in food packaging. Researchers have sought to enhance polysaccharide-based packaging through derivatization, nanocomposite preparation, and the incorporation of biological extracts.

Although polysaccharide-based packaging materials show a wide range of potential in areas such as food preservation, active packaging, and smart packaging, they still face challenges in practical applications such as cost, processing performance, and compatibility with existing production lines. In addition, natural polysaccharide-based packaging materials are moisture sensitive, possess poor mechanical properties, and are susceptible to microbial contamination, which limits their further development and application. Future research should focus on enhancing the performance of polysaccharide-based packaging materials, reducing costs, and exploring their large-scale applications in the food packaging industry.

In conclusion, polysaccharide-based materials hold significant potential for food packaging and, with continued research efforts, are likely to gradually replace plastic packaging and become widely adopted in the industry.

## Figures and Tables

**Figure 1 foods-13-03674-f001:**
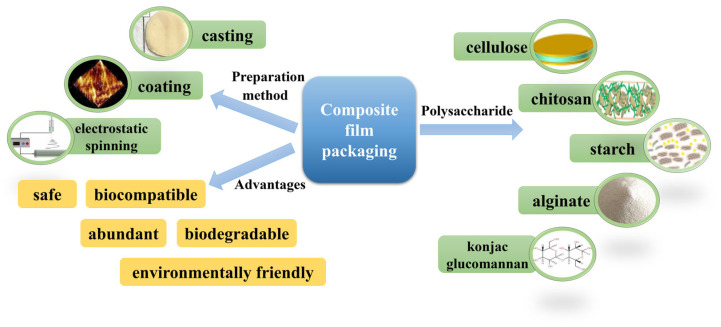
Preparation method and classification of polysaccharide-based composite films.

**Figure 3 foods-13-03674-f003:**
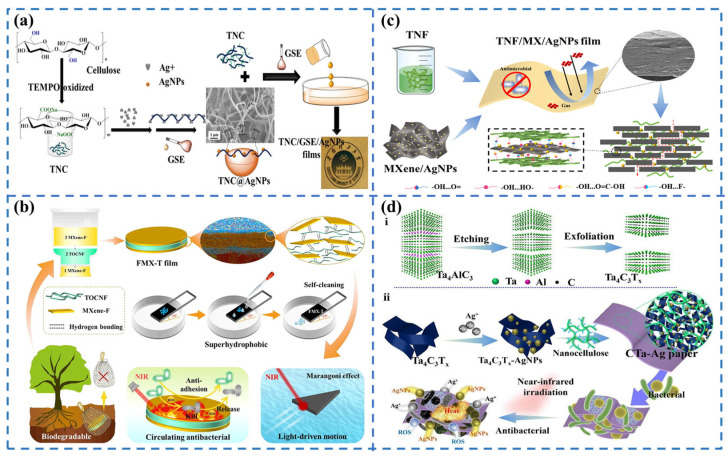
(**a**) Schematic diagram of the preparation of TNC/GSE/AgNPs film [34]; (**b**) Schematic preparation of a multifunctional sandwich-like composite membrane based on superhydrophobic MXene [35]; (**c**) Schematic preparation of biodegradable nanocellulose/MXene/AgNPs films [36]; (**d**) Schematic preparation of CTa-Ag packaging [37].

**Figure 5 foods-13-03674-f005:**
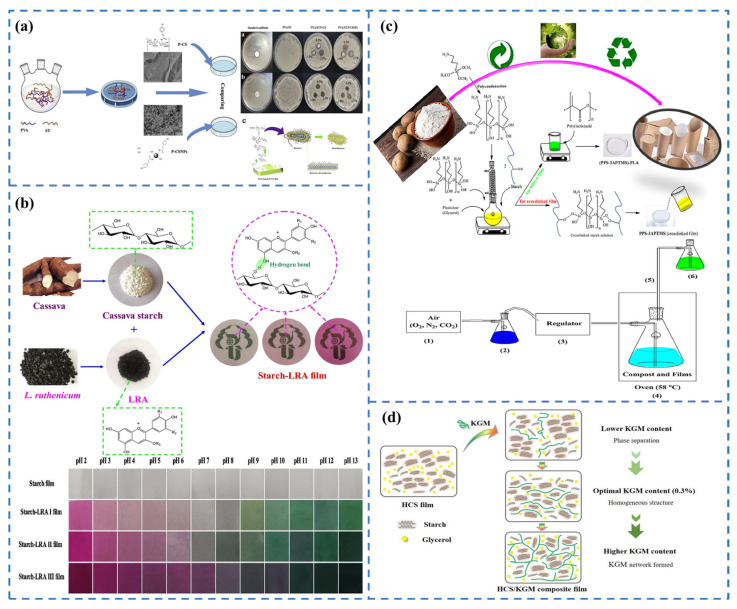
(**a**) Preparation and antimicrobial properties of chitosan nanoparticles/polyvinyl alcohol/starch composite films [56]; (**b**) Intermolecular interactions between starch and LRA and the monitoring of pork freshness by means of thin films [57]; (**c**) Schematic preparation of bilayer PPS-3APTMS-PLA film and biodegradation of the film [59]; (**d**) Schematic of HCS/KGM composite film development [60].

## Data Availability

The original contributions presented in the study are included in the article. Further inquiries can be directed to the corresponding author.

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
