# Peer review of "Polysaccharide-Based Composite Films: Promising Biodegradable Food Packaging Materials"

_foods, 2024, doi:10.3390/foods13223674_

Round 1

Reviewer 1 Report

Comments and Suggestions for Authors

Authors,

You have articulated your objectives for the Polysaccharide-based composite films: Promising biodegradable food packaging materials perfectly. The clarity and relevance of your aims set a strong foundation for the study. However, I would like to suggest the following corrections to enhance your results and discussion section:

Line 5. What is this e-mail@e-mail.com?

Abstract: Ok.

Introduction

Line 33-34. What do you mean by 'short'? Short compared to what? For instance, cellulose acetate can take more than three years to fully biodegrade.

Line 41. You already say degradable. Don’t repeat. 

You should rewrite this paragraph. Researchers have shifted their attention to biomass resources, using natural bio- 35

based polymers to prepare biodegradable packaging films instead of traditional plastic 36

ones. Natural biodegradable polymers, including polysaccharides (such as starch, cellu- 37

lose, chitosan, sodium alginate and hemicellulose), proteins (such as soy protein, collagen) 38

and lipids [7], are not only fully degradable but also inexpensive, biocompatible and suit- 39

able for preparing environmentally friendly packaging films. Therefore, their use in food 40

packaging films offers the advantages being green, safe and degradable. Because biomass 41

polysaccharides are abundant, biocompatible and biodegradable, they are considered 42

the most suitable raw materials for biodegradable packaging films. They not only provide 43

physical protection but can also be modified to impart specific functionalities such as an- 44

timicrobial, antioxidant and moisturizing properties [8]. This paragraph has some repetitions. 

Figure 1. You should improve the quality of this image.

2.1 You say right! Dispersion not solution. Thanks for.

Line 88-89 While casting methods significantly influence the thickness and structure of biodegradable polymer films, it is crucial to acknowledge that a variety of other factors also play a vital role. These include environmental conditions, such as room temperature, the specific solvent utilized, the type of polymer chosen, and the skill level of the individual preparing the film. Thus, the outcome is not determined solely by the casting method but rather by the intricate interplay of these multiple factors throughout the film-making process. If the successful production of biodegradable polymer films depended only on casting methods, addressing the associated challenges would be much simpler. So, rewrite this, or don’t write this. 

Line 90 – 97. Avoid using cellulose acetate in this context, as films produced via extrusion exhibit superior properties compared to those made through casting methods. Instead, consider utilizing alternative polymers that can provide better performance under casting conditions. In the context of your paper, cellulose acetate is not a good example. 

2.2 Line 103. `This method is cost-effective and the bio-based films can degrade naturally`. This is not accurate. Whether using the solvent casting method or the coating method, the biodegradability of the film depends primarily on the inherent properties of the polymer. If the polymer is biodegradable, the choice of method does not affect its ability to degrade. However, employing either method can enhance biodegradability by optimizing the film's structure and characteristics. Rewrite.

Line 129 In this study, FA and S with.. What is this?

2.3 Line 137 and 143. `polymer dispersion`.

Figure 2. Dear authors, I can`t read what is in the (a), (b),(d) and (e), improve this. Maybe you should divide it in more than one figure. 

3.1 `This approach enhances the material's mechanical strength of the material, while providing barrier and antimicrobial effects. ` Is not JUST about the approach that you said, is about more than this, for example, concentration of additives etc. Rewrite.  

214. The word `strong` is too much for that. Put another word. 

Line 229 Strong again. Put another word.

In this section I liked the way that you reference another author, start with the year and then say their names. Congrats. 

3.2. 259 `and forms a thin film` replace for `and forms a film`.

304- Film- forming dispersion not solution.

Fig 5. I can’t read what is inside the picture. Please, improve it.

Author Response

Reviewer #1: You have articulated your objectives for the Polysaccharide-based composite films: Promising biodegradable food packaging materials perfectly. The clarity and relevance of your aims set a strong foundation for the study. However, I would like to suggest the following corrections to enhance your results and discussion section:

Author reply: Thank you for your recognition of this manuscript and for your affirmation of our research work. We have reviewed and revised your suggestion, relevant revisions are presented as below.

  1. Line 5. What is this e-mail@e-mail.com?

Author reply: Thank you for your careful check. This was originally included in the template and we did not remove it. Now we have removed it.

Introduction

  1. Line 33-34. What do you mean by 'short'? Short compared to what? For instance, cellulose acetate can take more than three years to fully biodegrade.

Author reply: Thank you for your kind advice. Our intention was to compare the degradation time of bioplastics with that of petroleum-based plastics. Relevant revisions are presented as below:

Compared to petroleum-based plastics, which take up to 500 years or more to degrade, these materials can decompose naturally within a shorter period of time (several months to years) after use, thus reducing the impact on the environment.

  1. Line 41. You already say degradable. Don’t repeat.

Author reply: Thank you for your kind advice. We have deleted this sentence to minimize the use of “degradable”, while ensuring logical flow. The deleted sentence is as follows:

Because biomass polysaccharides are abundant, biocompatible and biodegradable, they are considered the most suitable raw materials for biodegradable packaging films.

4.You should rewrite this paragraph. Researchers have shifted their attention to biomass resources, using natural bio-based polymers to prepare biodegradable packaging films instead of traditional plastic ones. Natural biodegradable polymers, including polysaccharides (such as starch, cellulose, chitosan, sodium alginate and hemicellulose), proteins (such as soy protein, collagen) and lipids [7], are not only fully degradable but also inexpensive, biocompatible and suitable for preparing environmentally friendly packaging films. Therefore, their use in food packaging films offers the advantages being green, safe and degradable. Because biomass polysaccharides are abundant, biocompatible and biodegradable, they are considered the most suitable raw materials for biodegradable packaging films. They not only provide physical protection but can also be modified to impart specific functionalities such as antimicrobial, antioxidant and moisturizing properties [8]. This paragraph has some repetitions.

Author reply: Thank you for your kind suggestion. We have carefully reviewed and revised the paragraph. Relevant revisions are presented as below:

Researchers have shifted their attention to biomass resources, using natural bio-based polymers to prepare biodegradable packaging films instead of traditional plastic ones. Natural polymers, including polysaccharides (such as starch, cellulose, chitosan, sodium alginate and hemicellulose), proteins (such as soy protein, collagen) and lipids [1], are not only fully degradable but also inexpensive and biocompatible. Therefore, their use in food packaging films offers the advantages being green, safe and degradable. In addition, they not only provide physical protection but can also be modified to impart specific functionalities such as antimicrobial, antioxidant and moisturizing properties [2].

  1. Figure 1. You should improve the quality of this image.

Author reply: Thank you for your kind suggestion. We've improved figure 1. Relevant revisions are presented as below:

Figure 1. Preparation method and classification of polysaccharide based composite films.

  1. Line 88-89 While casting methods significantly influence the thickness and structure of biodegradable polymer films, it is crucial to acknowledge that a variety of other factors also play a vital role. These include environmental conditions, such as room temperature, the specific solvent utilized, the type of polymer chosen, and the skill level of the individual preparing the film. Thus, the outcome is not determined solely by the casting method but rather by the intricate interplay of these multiple factors throughout the film-making process. If the successful production of biodegradable polymer films depended only on casting methods, addressing the associated challenges would be much simpler. So, rewrite this, or don’t write this.

Author reply: Thank you for your kind advice. We've removed the sentence to ensure accuracy of expression. The removed sentence is as follows:

The casting method allows for precise control of the film's thickness and structure, ensuring sufficient strength while maintaining flexibility.

  1. Line 90 – 97. Avoid using cellulose acetate in this context, as films produced via extrusion exhibit superior properties compared to those made through casting methods. Instead, consider utilizing alternative polymers that can provide better performance under casting conditions. In the context of your paper, cellulose acetate is not a good example.

Author reply: Thank you for your kind advice. We have carefully reviewed and revised the example. Relevant revisions are presented as below:

For example, Othman, SH et al.[3] developed corn starch/chitosan nanoparticles/thymol (CS/CNP/Thy) bio-nanocomposite films as potential food packaging materials that can enhance the shelf life of food. CS/CNP/Thy bio-nanocomposite films were prepared by the addition of different concentrations of thymol (0, 1.5, 3.0, 4.5 w/w%) using a solvent casting method. It was found that although the addition of thymol reduced the mechanical properties of the film, it was effective in lengthening the shelf life of cherry tomatoes, maintaining their firmness, reducing weight loss and inhibiting mold growth by adjusting the concentration of thymol, especially at 3%. This suggests that CS/CNP/Thy bio-nanocomposite films have the potential to be applied in the field of active food packaging to improve the preservation quality and prolong the shelf life of food.

  1. 2.2 Line 103. `This method is cost-effective and the bio-based films can degrade naturally`. This is not accurate. Whether using the solvent casting method or the coating method, the biodegradability of the film depends primarily on the inherent properties of the polymer. If the polymer is biodegradable, the choice of method does not affect its ability to degrade. However, employing either method can enhance biodegradability by optimizing the film's structure and characteristics. Rewrite.

Author reply: Thank you for your kind suggestion. We have removed this sentence and the advantages of coating are detailed in the last paragraph of this section. Relevant revisions are presented as below:

The coating method is generally simple and easy to implement and does not re-quire complex equipment. However, spraying requires a safe and non-toxic film-forming solution, and the sprayed film can be unevenly distributed, which limits its scope of application.

  1. Line 129 In this study, FA and S with What is this?

Author reply: Thank you for your kind advice. FA is fulvic acid and S is sericin. We have revised and explained in the review.

  1. 2.3 Line 137 and 143. `polymer dispersion`.

Author reply: Thank you for your question. We have modified the phrase.

  1. Figure 2. Dear authors, I can`t read what is in the (a), (b), (d) and (e), improve this. Maybe you should divide it in more than one figure.

Author reply: Thank you for your kind advice. There is no (e) diagram in Figure 2. (a) diagram shows the flow chart of polymer films preparation by casting method. (b) diagram shows the fabrication route of CNCs and PLA composites. (d) diagram shows the paper packaging material prepared by electrospray method to extend the shelf life of pears.

  1. 3.1 `This approach enhances the material's mechanical strength of the material, while providing barrier and antimicrobial effects. ` Is not JUST about the approach that you said, is about more than this, for example, concentration of additives etc. Rewrite.

Author reply: Thank you for your kind suggestion. We have carefully reviewed and revised the sentence. Relevant revisions are presented as below:

In recent years, increasing numbers of researchers have shown interest in composites prepared from cellulose nanopaper or recycled cellulose films with certain organic, inorganic or natural additives such as nano-silver, nano-silicon dioxide, chitosan, and plant essential oils, which enhances the material's mechanical strength of the material, while providing barrier and antimicrobial effects.

  1. 214. The word `strong` is too much for that. Put another word.

Author reply: Thank you for your careful check. We have carefully revised the sentence. Relevant revisions are presented as below:

In addition, the film showed strong antioxidant and antimicrobial activity. Achieving super mechanical strength and high barrier properties continues to be a significant challenge.

  1. Line 229 Strong again. Put another word.

Author reply: Thank you for your kind advice. We have carefully reviewed and revised the manuscript. Replacements are marked red in the article.

15 3.2. 259 `and forms a thin film` replace for `and forms a film`.

Author reply: Thank you for your careful check. We have carefully revised the sentence. Relevant revisions are presented as below:

In a mildly acidic environment, chitosan dissolves and forms a film.

  1. 304- Film- forming dispersion not solution.

Author reply: Thank you for your careful check. We have carefully revised the sentence. Relevant revisions are presented as below:

Metal nanoparticles currently used in food packaging include nano-copper[4], nano-silver[5], and some metal oxides such as zinc oxide nanoparticles[6] and nano-titanium dioxide[7], which are often added to the film forming dispersion to enhance the properties of the chitosan-based film, such as the mechanical, barrier, thermal, optical, and anti-microbial properties.

  1. Fig 5. I can’t read what is inside the picture. Please, improve it.

Author reply: Thank you for your kind advice. The figures in this article have been resized in accordance with the layout, and they are all high-definition figures that can be read clearly; the figures of the review articles are provided to enable readers to better understand the general content of the articles, so as to understand the intent of our examples.

  1. Mohamed, S. A. A.; El-Sakhawy, M.; El-Sakhawy, M. A.-M., Polysaccharides, Protein and Lipid -Based Natural Edible Films in Food Packaging: A Review. Carbohydrate Polymers 2020, 238.
  2. Wang, Y.; Liu, K.; Zhang, M.; Xu, T.; Du, H.; Pang, B.; Si, C., Sustainable polysaccharide-based materials for intelligent packaging. Carbohydrate Polymers 2023, 313.
  3. Othman, S. H.; Othman, N. F. L.; Shapi'i, R. A.; Ariffin, S. H.; Yunos, K. F. M., Corn Starch/Chitosan Nanoparticles/Thymol Bio-Nanocomposite Films for Potential Food Packaging Applications. Polymers 2021, 13 (3).
  4. Jayaramudu, T.; Varaprasad, K.; Pyarasani, R. D.; Koteshwara Reddy, K.; Dileep Kumar, K.; Akbari-Fakhrabadi, A.; Mangalaraja, R. V.; Amalraj, J., Chitosan capped copper oxide/copper nanoparticles encapsulated microbial resistant nanocomposite films. International Journal of Biological Macromolecules 2019, 128, 499-508.
  5. Affes, S.; Maalej, H.; Aranaz, I.; Kchaou, H.; Acosta, N.; Heras, A.; Nasri, M., Controlled size green synthesis of bioactive silver nanoparticles assisted by chitosan and its derivatives and their application in biofilm preparation. Carbohydrate Polymers 2020, 236.
  6. Roy, S.; Priyadarshi, R.; Rhim, J.-W., Development of Multifunctional Pullulan/Chitosan-Based Composite Films Reinforced with ZnO Nanoparticles and Propolis for Meat Packaging Applications. Foods 2021, 10 (11).
  7. Roy, S.; Zhai, L.; Kim, H. C.; Pham, D. H.; Alrobei, H.; Kim, J., Tannic-Acid-Cross-Linked and TiO2-Nanoparticle-Reinforced Chitosan-Based Nanocomposite Film. Polymers 2021, 13 (2).

Reviewer 2 Report

Comments and Suggestions for Authors

Dear Authors,

In general terms, a better drafting of the manuscript is recommended. Colloquial expressions were observed that do not correspond to a scientific article. It is important to point out to the authors that they are not publishing in a popular journal, but in a scientific one.

The observations in their manuscript are marked in yellow. Some sentences and words lack clarity, vocabulary, sentence consistency, and punctuation marks, among other grammatical writing errors.

Figure 1 is minimal; it only mentions three polysaccharides when there are more than 10 polysaccharides used as recurrences or compound films. This figure should be improved with the correct information.

The figures in this article are a compilation of images of the references, with various sizes that make them unreadable. It is not possible to have copies of the graphical abstracts for a review article with an improvement in the quality of these images.

The scientific names of the microorganisms should be written in italics.

In my opinion, the manuscript has the potential to provide valuable insights to the field. However, it currently lacks a thorough analysis of the information provided by the authors. A substantial change in the information provided is required for its evaluation and to indicate a review with sufficient scientific quality.

Dozens of polysaccharides are applied for composite films and edible coatings. The authors rely on three polysaccharides (cellulose, chitosan, and starch) and only devote section 3.4 to other polysaccharides. 

The document requires a complete revision of the English language due to grammatical errors that have been pointed out. A native English speaker should review this document. Authors are requested to avoid the use of AI programs for English editing.

Yours sincerely,

The reviewer.

Comments on the Quality of English Language

The document requires a complete revision of the English language due to grammatical errors that have been pointed out. A native English speaker should review this document. Authors are requested to avoid the use of AI programs for English editing.

Author Response

Dear editors and reviewers:

Thanks for your comments concerning our manuscript entitled “Polysaccharide-based composite films: Promising biodegradable food packaging materials”. Those comments are valuable and very helpful for revising and improving our manuscript. We have carefully made corrections according to the suggestion. The revised parts are marked in red in the manuscript. The main corrections in the manuscript and the response to reviewers’ comments are as follows:

Reviewer: In general terms, a better drafting of the manuscript is recommended. Colloquial expressions were observed that do not correspond to a scientific article. It is important to point out to the authors that they are not publishing in a popular journal, but in a scientific one.

Author reply: Thank you for your patient and meticulous review of the manuscript. You have provided many valuable suggestions for paper revision, and all of your suggestions are crucial for manuscript revision. We have reviewed and revised your suggestions, relevant revisions are presented as below.

  1. The observations in their manuscript are marked in yellow. Some sentences and words lack clarity, vocabulary, sentence consistency, and punctuation marks, among other grammatical writing errors.

Author reply: Thank you for your kind advice. We have carefully reviewed and revised the manuscript.

  1. Figure 1 is minimal; it only mentions three polysaccharides when there are more than 10 polysaccharides used as recurrences or compound films. This figure should be improved with the correct information.

Author reply: Thank you for your kind advice. We've improved figure 1. Relevant revisions are presented as below:

Figure 1. Preparation method and classification of polysaccharide based composite films.

  1. The figures in this article are a compilation of images of the references, with various sizes that make them unreadable. It is not possible to have copies of the graphical abstracts for a review article with an improvement in the quality of these images.

Author reply: Thank you for your kind suggestion. The figures in this article have been resized in accordance with the layout, and they are all high-definition figures that can be read clearly; the figures of the review articles are provided to enable readers to better understand the general content of the articles, so as to understand the intent of our examples. In our opinion, our figures are reasonable.

  1. The scientific names of the microorganisms should be written in italics.

Author reply: Thank you for your kind advice. We have carefully checked the manuscript and revised relevant content in the manuscript. Replacements are marked red in the article.

  1. Dozens of polysaccharides are applied for composite films and edible coatings. The authors rely on three polysaccharides (cellulose, chitosan, and starch) and only devote section 3.4 to other polysaccharides.

Author reply: Thank you for your kind advice. More and more polysaccharides are applied to food packaging composite films, but our article focuses on three polysaccharides with the widest application, the best effect, and the most researches to summarize and look forward. If all the polysaccharides are introduced one by one, it will lead to the problems of redundancy of content and lack of focus, so we mainly introduce three polysaccharides.

  1. The document requires a complete revision of the English language due to grammatical errors that have been pointed out. A native English speaker should review this document. Authors are requested to avoid the use of AI programs for English editing.

Author reply: Thank you for your kind advice. We've rechecked the whole article and fixed the errors.

Reviewer 3 Report

Comments and Suggestions for Authors

This review entitled "Polysaccharide-based composite films: Promising biodegradable food packaging materials" is a work that deals with some important aspects of the synthesis of chitosan, cellulose and starch films. It provides relevant aspects of synthesis types such as casting, electrospinning and coating. It is a complete work, but the authors should elaborate on two important aspects which are 1-biocompatibility and 2-biodegradability separately in the form of subtitles in order to make the review more prominent.

Comments on the Quality of English Language

I´m not qualified.

Author Response

Dear editors and reviewers:

Thanks for your comments concerning our manuscript entitled “Polysaccharide-based composite films: Promising biodegradable food packaging materials”. Those comments are valuable and very helpful for revising and improving our manuscript. We have carefully made corrections according to the suggestion. The revised parts are marked in red in the manuscript. The main corrections in the manuscript and the response to reviewers’ comments are as follows:

Reviewer: This review entitled "Polysaccharide-based composite films: Promising biodegradable food packaging materials" is a work that deals with some important aspects of the synthesis of chitosan, cellulose and starch films. It provides relevant aspects of synthesis types such as casting, electrospinning and coating. It is a complete work, but the authors should elaborate on two important aspects which are 1-biocompatibility and 2-biodegradability separately in the form of subtitles in order to make the review more prominent.

Author reply: Thank you for taking the time to read and provide me with suggestions for revisions to improve the quality of our manuscript. Regarding the biocompatibility and biodegradability of polysaccharide composite films as the main advantages of polysaccharide composite films, we have already elaborated on them several times in the sections of the article such as introduction and applications, and have explained them in detail in the process of giving examples. In our review, adding further subsections for separate elaboration would result in repetition of the contents of the article.

Reviewer 4 Report

Comments and Suggestions for Authors

The authors of the manuscript foods-3288632-peer-review-v1, presented review article on Polysaccharide-based composite and their potential use for food packaging industry. The topic is interesting and suited for the journal interest. However, the manuscript needs to be improved and some points need to be clarified.

-        There is a gap in the literature review concerning the current research in the Polysaccharide-based composite. Authors are invited to present a chronological development of background, current research, gaps, and necessity of this research.

-        Section 2, revised to “Preparation method of Polysaccharide-based packaging films”

-        Sec 3 that entitled “Applications for polysaccharide-based food packaging film” the content of that section needs to be re-construct and divided into various subsection and include additional reviews. For example, authors added a separate section entitled “Polysaccharide-based Film properties” and reviewed for the barrier properties (Water vapour permeability) and mechanical properties of pure polysaccharides or blend of polysaccharides”. Lastly, a separate section for the polysaccharides film applications including challenges and perspectives and regulatory and safety issues.

Author Response

Dear editors and reviewers:

Thanks for your comments concerning our manuscript entitled “Polysaccharide-based composite films: Promising biodegradable food packaging materials”. Those comments are valuable and very helpful for revising and improving our manuscript. We have carefully made corrections according to the suggestion. The revised parts are marked in red in the manuscript. The main corrections in the manuscript and the response to reviewers’ comments are as follows:

Reviewer: The authors of the manuscript foods-3288632-peer-review-v1, presented review article on Polysaccharide-based composite and their potential use for food packaging industry. The topic is interesting and suited for the journal interest. However, the manuscript needs to be improved and some points need to be clarified.

Author reply: Thank you for your patient and meticulous review of the manuscript. You have provided many valuable suggestions for paper revision, and all of your suggestions are crucial for manuscript revision. We have reviewed and revised your suggestions, relevant revisions are presented as below.

  1. There is a gap in the literature review concerning the current research in the Polysaccharide-based composite. Authors are invited to present a chronological development of background, current research, gaps, and necessity of this research.

Author reply: Thank you for your kind advice. This review focuses on the application of polysaccharide-based packaging films in food preservation. We first discussed the preparation methods of polysaccharide-based packaging films and their advantages and disadvantages. Next, we discussed in depth the research and application progress of several polysaccharide-based composite films. Finally, we summarized and discussed the challenges faced by polysaccharides and the future development direction in the field of food packaging films. To our knowledge, the research background, development overview, and research significance of polysaccharide-based composites have been covered in reviews, so we focus on their preparation and application aspects to complement them.

  1. Section 2, revised to “Preparation method of Polysaccharide-based packaging films”

Author reply: Thank you for your careful check. We've revised the title.

  1. Sec 3 that entitled “Applications for polysaccharide-based food packaging film” the content of that section needs to be re-construct and divided into various subsection and include additional reviews. For example, authors added a separate section entitled “Polysaccharide-based Film properties” and reviewed for the barrier properties (Water vapour permeability) and mechanical properties of pure polysaccharides or blend of polysaccharides”. Lastly, a separate section for the polysaccharides film applications including challenges and perspectives and regulatory and safety issues.

Author reply: Thank you for your kind advice. In the “Applications for polysaccharide-based food packaging films” section, we have provided detailed descriptions of the applications of each of the three polysaccharide-based composite films, with a review in each subsection. The idea is to follow the applications of the different polysaccharide-based films, and each section contains reviews of barrier and mechanical properties, so we have not added separate reviews for these two sections. In the “Summary and outlook” section, we discussed the challenges faced by polysaccharide-based films as well as their development prospects, so they are not discussed separately in the third section.

Round 2

Reviewer 1 Report

Comments and Suggestions for Authors

The authors followed my recommendations. Thank you!

Author Response

The authors followed my recommendations. Thank you!

Author reply: Thank you for your help and affirmation. Thank you again for your positive comments and valuable suggestions to improve the quality of our manuscript.

Reviewer 2 Report

Comments and Suggestions for Authors

Dear Authors,

The comments I have on your document are as follows:

1) You should have addressed all the points that were marked in your manuscript of the first version.

2) There is a lack of punctuation marks throughout your manuscript.

3) The texts that are written in red color are texts that were already written in the first version; in fact, it can be noticed that the others are marked complete paragraphs.

4) Authors are asked to mark the modified words or sentences in different colored text.

I consider that the document should be revised again and adapted to these provisions.

The reviewer.

Comments on the Quality of English Language

I suggest a revision of the punctuation marks in the manuscript.

Author Response

Dear editors:

Thanks for your comments concerning our manuscript entitled “Polysaccharide-based composite films: Promising biodegradable food packaging materials”. Those comments are valuable and very helpful for revising and improving our manuscript. We have carefully made corrections according to the suggestion. The revised parts are marked in red in the manuscript. The main corrections in the manuscript and the response to reviewers’ comments are as follows:

The comments I have on your document are as follows:

1) You should have addressed all the points that were marked in your manuscript of the first version.

Author reply: Thank you for your kind advice. After two revisions, we have modified all the points that were marked in the manuscript of the first version.

2) There is a lack of punctuation marks throughout your manuscript.

Author reply: Thank you for your kind suggestion. We have carefully reviewed and revised the manuscript. We've checked all the punctuation and added all the missing punctuation marks.

3) The texts that are written in red color are texts that were already written in the first version; in fact, it can be noticed that the others are marked complete paragraphs.

Author reply: Thank you for your kind advice. In the revised manuscript, we have marked the sentences with changes in red, even though there were some sentences where only words or punctuation had been changed, resulting in the red range of the markers to be too large. So it might have been easy to mistakenly assume that no changes had been made. We have improved this issue by red-highlighting only the changes.

4) Authors are asked to mark the modified words or sentences in different colored text.

Author reply: Thank you for your kind advice. We have marked all the revision in red and have not indicated them in any other color, but we have carefully reviewed and revised the manuscript and corrected all the issues in accordance with the review comments.

Reviewer 4 Report

Comments and Suggestions for Authors

The revised manuscript has been improved. Authors responded to all comments, and I accepted their answers. Hence, I recommend the current version of the manuscript for publication in Foods

Author Response

I consider that the document should be revised again and adapted to these provisions.

The revised manuscript has been improved. Authors responded to all comments, and I accepted their answers. Hence, I recommend the current version of the manuscript for publication in Foods.

Author reply: Thank you for your help and affirmation. Thank you again for your positive comments and valuable suggestions to improve the quality of our manuscript.